# Neurosurgeons' experiences of conducting and disseminating clinical research in low- and middle-income countries: a qualitative study protocol

Charlotte J Whiffin [1,2,3] Brandon G Smith [2,3] Ignatius N Esene,[3,4] Claire Karekezi,[5] Tom Bashford [2,3] Muhammad Mukhtar Khan,[6] Davi J Fontoura Solla,[7] Peter J Hutchinson,[2,3] Angelos Kolias[2,3]

CJW and BGS are joint first authors.

For numbered affiliations see end of article.

**Correspondence to**
Dr Charlotte J Whiffin;
c.whiffin@derby.ac.uk

## ABSTRACT

**Introduction** Low-income and middle-income countries (LMICs) face the greatest burden of neurotrauma. However, most of the research published in scientific journals originates from high-income countries, suggesting those in LMICs are either not engaging in research or are not publishing it. Evidence originating in high-income countries may not be generalisable to LMICs; therefore, it is important to nurture research capacity in LMICs so that a relevant evidence base can be developed. However, little is published about specific challenges or contextual issues relevant to increasing research activity of neurosurgeons in LMICs. Therefore, the aim of this study was to understand neurosurgeons' experiences of, aspirations for and ability to conduct and disseminate clinical research in LMICs.

**Methods and analysis** This is a pragmatic qualitative study situated within the naturalistic paradigm using focus groups and interviews with a purposive sample of neurosurgeons from LMICs. First, we will conduct asynchronous online focus groups with 36 neurosurgeons to broadly explore issues relevant to the study aim. Second, we will select 20 participants for follow-up semistructured interviews to explore concepts in more depth and detail than could be achieved in the focus group. Interviews will be audio-recorded and transcribed verbatim. A thematic analysis will be conducted following Braun and Clarke's six stages and will be supported by NVIVO software.

**Ethics and dissemination** The University of Cambridge Psychology Research Ethics Committee reviewed this study and provided a favourable opinion in January 2020 (REF PRE.2020.006). Participants will provide informed consent, be able to withdraw at any time and will have their contributions kept confidential. The findings of the study will be shared with relevant stakeholders and disseminated in conference presentations and journal publications.

## Strengths and limitations of this study

► First qualitative study to explore the challenges of neurosurgeons conducting and disseminating clinical research in low-income and middle-income countries (LMICs).

► Provides an in-depth exploration of neurosurgeons' understanding of how engagement in research could be improved.

► Knowledge generated from this study will inform recommendations to enhance the research capacity of neurosurgeons in LMICs.

► It will not be possible to represent all LMICs in this study.

► Excluding non-English-speaking neurosurgeons will mean any additional barriers that these neurosurgeons may face in conducting and disseminating research will remain unexplored.

## INTRODUCTION

In 2017, the UK Department of Health and Social Care provided over £160 million of funding for global health research.[1] From this funding, the National Institute for Health Research (NIHR) Global Health Research Group on Neurotrauma was established with the overall aim of improving outcomes from neurotrauma in low-income and middle-income countries[2] (LMICs). In addition to improving outcomes, the group also aims to increase the participation of LMICs in high-quality clinical research.[3] The low proportion of published scientific papers from authors in LMICs is stark.[4 5] The limited participation of LMICs in research contrasts sharply with their disproportionately high incidence of neurotrauma.[6] In a recent study of research productivity in LMICs, as evidenced by publication in journals, Servadei *et al*[5] found that only 4.52% of 6708 published reports had an LMIC affiliation. In a further bibliometric study, Africa and Southeast Asia were found to be responsible for less than 3% of publication productivity.[7] If journal publication is a reliable indicator of the quality and quantity of research, then LMICs are severely underrepresented in an international context.[5] In

addition, Kolias *et al*[3] asserts that multicentre trials are typically conducted in high-income countries, making generalisability of these studies to environments with different treatment practices particularly problematic. A recent review to identify randomised trials of neurosurgical procedures used in cranial and spinal neurosurgical practice found only 8.8% of studies (excluding China) were LMIC-led studies.[8] LMICs need to develop an evidence base that is relevant to the treatments and interventions accessible to them.[3 7] Nurturing an environment that promotes high-quality neurotrauma research is a fundamental part of this.

Little is written in the academic literature about the reasons why there are so few studies conducted, and/or published, by neurosurgeons in LMICs, although lack of time and resources are a likely factor.[5] More specifically, Langer *et al*[4] suggested poor research production; poor preparation of manuscripts; poor access to scientific literature; poor participation in publication-related decision-making processes and bias of journals all exacerbate low engagement. However, there is little empirical data to underpin these assumptions; therefore, it is important to understand this so that any recommendations for enhancing research capacity in the future are context specific and borne out of an in-depth understanding of the problem. A review by Franzen *et al*[9] examined health research capacity development in LMICs suggesting power relationships effected capacity development, that stronger links between research, policy and practice were required and that a systems response was necessary if capacity was to be improved. However, only 20.8% of the papers included in the review were sourced from empirical primary research.

### Aim and objectives

The aim of this study is to understand neurosurgeons' experiences of, aspirations for and ability to conduct and disseminate clinical research in LMICs.

### Objectives

1. Explore the types of clinical research neurotrauma physicians from LMICs are engaged in.
2. Understand the contextual challenges to conducting and disseminating clinical research in LMICs.
3. Identify ways in which research and dissemination activities can be facilitated in LMICs.

## METHODS AND ANALYSIS
### Study design

We propose an exploratory pragmatic qualitative study situated within the naturalistic paradigm. Pragmatic qualitative research, also referred to as descriptive or generic qualitative research, is particularly useful when little is known about a topic and when researching populations from other cultures.[10] Pragmatic qualitative studies are not philosophically informed, allowing the study to be designed in a way that is feasible, achievable and appropriate for the aim of the study.[11] The naturalistic paradigm,

explicated by Lincoln and Guba,[12] also known as the constructivist paradigm in more recent texts (although many authors still prefer this original description), reject methods which are reductionistic.[13] Naturalistic inquiry tries to stay true to the nature of the phenomena under investigation and commits to the existence of multiple realities and working with subjectivity.[14]

### Data collection

As there is little written in the empirical literature, we were cautious about developing an a priori interview schedule to guide data collection. Therefore, we have designed a study that will explore neurosurgeons' experiences within a focus group first as a means to understand the nature of the problem initially and then conduct semistructured interviews with a smaller sample to explore the research question in more depth than may possible in a focus group setting. In addition, we propose collecting a small amount of demographic data.

Specific demographic data will be collected from each participant to include age, gender, country of residence, job title and experience. In addition, we will ascertain specific exposure to research training, research participation and the relevance of research to job role and job progression. These data are required if we are to interpret the context of the data correctly.

Online focus groups have a number of advantages over more traditional focus group formats.[15] First, they allow people who are in different geographical places to contribute to a group discussion where a face-to-face focus group is not possible.[16] Second, there is a heightened level of anonymity which may mean people feel more able to be honest and or share sensitive information. Third, data are immediately available without the necessity for transcription.[15] In addition, the asynchronous format allows participants the ability to contribute at a convenient place and time, making the research more accessible to participants who may be in different time zones with competing demands on their time.[17]

The focus group questions were discussed with members of the research team, particularly those from LMICs, to ensure these were appropriate (box 1).

Selection of the online platform to facilitate the focus group was informed by a number of strict criteria informed by principles of safe data storage and accessibility. Each

---

> **Box 1    Focus group questions**
>
> 1. What are your personal experiences of conducting clinical research and what personal and/or organisational factors motivate you to conduct research?
> 2. What specific barriers are there to you conducting clinical research within your hospital?
> 3. In what ways is research shared between colleagues, the public and the wider academic community?
> 4. What would help you to conduct and publish good clinical research?
> 5. What unique factors are there that should be considered to nurture research capacity in low-income and middle-income countries?

participant will be given a unique username and individually assigned password allowing them to contribute anonymously to the focus group. We propose three separate focus groups: Group 1 (lower income); Group 2 (lower-middle income); Group 3 (upper middle income). We will try to have representation from a wide range of countries but will be affected by the number of people who respond to the initial call for participants. Questions will be posted online every 7–10 days in the order listed. Members of the UK research team will regularly check the forum during this time, respond to direct questions if necessary, ask additional follow-up questions where appropriate or request clarification of points made.

Following completion of the online focus groups, we will invite 20 participants to complete online video or telephone semistructured interviews. We will select participants based on their demographic data and their contribution to the online focus group to ensure we capture a range of views and experiences. The specific interview schedule will be developed following preliminary analysis of the focus group data. The interviews will last approximately 30–60 min, will be conducted by the UK-based research team and will be recorded with a digital recorder.

## Sample
Non-random sampling is used in qualitative studies and here we use a purposive approach. Persons eligible to participate in this study are neurosurgeons working in a country defined as low-income or middle-income, self-declared fluency in written and spoken English, have access to, and able to use, a personal computer or smart phone and able to provide informed consent. Participants will be identified in countries participating in the NIHR Global Health Research Group on Neurotrauma listed in box 2. However, new collaborators in the group, including Zimbabwe and the Philippines, and participants from institutions based in other LMICs will be added if necessary to achieve the required sample size.

Qualitative studies do not make any claims about generalisability so a sample size calculation is not appropriate. Instead, qualitative studies use the concept of data

---

**Box 2  Low-income and middle-income countries in the National Institute for Health Research Global Health Research Group on Neurotrauma**

1. Brazil
2. Colombia
3. Ethiopia
4. India
5. Indonesia
6. Malaysia
7. Myanmar
8. Nigeria
9. Pakistan
10. South Africa
11. Tanzania
12. Zambia

---

saturation to assess the completeness of findings which is the point reached when researchers are confident that new data will reveal no new information.[18 19] In this study, we aim to recruit 36 participants for the online focus groups and then 20 participants for the individual semistructured follow-up interviews when we expect data saturation to be reached. If saturation is not reached, and increasing the sample size is feasible, we will request approval from the ethics committee to increase the sample to a more appropriate size.

## Data analysis
There will be three phases to analysis in this study mirroring the stages outlined under study design. Phase I will analyse the focus group data; phase II will analyse the semistructured interview data; and phase III will triangulate findings from all methods to determine final findings for the study.

Phases I and II will use a Braun and Clarke[20] thematic analysis (see box 3) which is commonly used in pragmatic qualitative studies[20 21] that do not require 'highly abstract rendering of data'[22p.3]. Audio files will be transcribed verbatim by a transcription service and checked for accuracy by the research team. Online focus groups will be downloaded from the online platform and transferred to a Microsoft Word file. Given the nature of the research question, it would be wrong to enforce an a priori framework on the analysis. Analysis will therefore be inductive and In Vivo coding will be used. Analysis will be supported by the use of NVIVO software allowing researchers to organise the data, share coding decisions and confirm the origins of interpretation. CW will lead on the analysis, supported by BS in the initial coding and exploration of the data. CW is an experienced qualitative researcher and nurse academic with clinical experience of neurosurgery. BS is a PhD student and fourth year medical student who has undertaken training in qualitative methods and analysis. Once initial themes are identified, these will be discussed with coauthors alongside supporting data. Themes will then be returned to participants so they can provide further insight. Any comments will be built into the process of defining and naming final themes.

Phase III will involve triangulation of data obtained from interviews and focus groups. The benefits of data triangulation include developing a more comprehensive understanding of the phenomena under investigation through the use of multiple methods.[23]

## Rigour
This study uses multiple methods and triangulation to increase its depth and accuracy of understanding. Preliminary findings from the focus group will be explored within the individual interviews which will increase credibility of the final findings. Critical reflexivity will also safeguard against naïve assumptions and possible hidden biases within the analysis. Peer debriefing and respondent validation will also be used in this study to increase rigour.

## Box 3 Braun and Clarke thematic analysis framework

1. Familiarising yourself with your data
2. Generate initial codes
3. Searching for themes
4. Reviewing themes
5. Defining and naming themes
6. Produce the report

## ETHICS AND DISSEMINATION

The University of Cambridge Psychology Research Ethics Committee reviewed this study in January 2020 (REF PRE.2020.006). The University of Cambridge is the sponsor and appropriate insurance is in place. Participants will be informed fully about the study methods, risks and benefits through the participant information sheet. A cooling off period of 48 hours will be given to all those who express an interest in the study and then electronic consent will be taken. Participants will be able to withdraw at any time. Data cannot be withdrawn from the focus groups; however, participants can request their data are withdrawn from the interviews for up to 1 week following their completion. After this period, these data cannot be withdrawn as analysis will have commenced.

All information will be kept strictly confidential and comply to principles of UK data protection law and General Data Protection Regulation. Participants will be advised not to share any personal information in the online focus group. When the focus groups have been completed, we will anonymise the data prior to data analysis. Similarly, all interview data will be anonymised on transcription. Participants will be told that anonymised quotes will be published in the findings of this study. We will also be seeking consent to disclose region and level of income associated with their country of origin against any quotes used in the publication of findings as this will be important to the contextual understanding of the study.

## Patient and public involvement

Our research question asks for the views of neurosurgeons and therefore we do not intend to include patients or the public in the design of, or data collection for, this study. However, we did ask for peer review of the study by collaborating members of the Gobal Health Research Group on Neurotrauma. Their comments informed the final study design.

## Dissemination

Participants in this study will be sent a summary of the findings once analysis has been completed. The findings of the study will then be shared through the NIHR GHRGN network, and other relevant stakeholders, including the World Federation of Neurosurgical Societies. This study will also be disseminated in conference presentations and journal publications.

## Study limitations

Generalisibility of the findings may be limited, as we do not expect to have participating neurosurgeons from all LMICs. However, we hope to include participants from a variety of LMICs with broad spread in terms of geography and income status. Unfortunately, due to the resources available, we had to exclude non-English-speaking neurosurgeons. Any additional experiences that this specific population have in conducting and disseminating research will remain unexplored; therefore, we will need to be cautious in our final conclusions for this study. However, given that this is the first study, that we know of, to explore the challenges of neurosurgeons conducting and disseminating clinical research in LMICs, this qualitative study will provide a rich and in-depth understanding of how engagement in research could be improved for this population. This understanding will facilitate the development of appropriate recommendations with the aim of nurturing research capacity for neurosurgeons in LMICs in the future.

**Author affiliations**
[1]College of Health and Social Care, University of Derby, Derby, UK
[2]Division of Neurosurgery, Department of Clinical Neurosciences, Addenbrooke's Hospital and University of Cambridge, Cambridge, UK
[3]NIHR Global Health Research Group on Neurotrauma, University of Cambridge, Cambridge, UK
[4]Neurosurgery Division, Faculty of Health Sciences, University of Bamenda, Bambili, NW Region, Cameroon
[5]Department of Neurosurgery, Rwanda Military Hospital, Kigali, Kigali City, Rwanda
[6]Neurosurgery, Northwest School of Medicine and Northwest General Hospital and Research Centre, Peshawar, Pakistan
[7]Department of Neurology, Division of Neurosurgery, University of Sao Paulo, Sao Paulo, Brazil

**Contributors** PH and AK conceived the research idea. CW, AK and BGS designed the study. CW wrote the first draft of the protocol and paper for publication. TB, INE, CK, MMK and DJFS all contributed to the study protocol and edited the final paper.

**Funding** This work was supported by the National Institute for Health Research Global Health Research Group on Neurotrauma.

**Competing interests** AK and PH are supported by the National Institute for Health Research (NIHR) Cambridge Biomedical Research Centre and the NIHR Global Health Research Group on Neurotrauma. PH is also supported by a NIHR Research Professorship and the Royal College of Surgeons of England. The NIHR Global Health Research Group on Neurotrauma was commissioned by the United Kingdom NIHR using Official Development Assistance funding (Project No. 16/137/105). The views expressed in this manuscript are those of the authors and are not necessarily those of the United Kingdom National Health Service, NIHR or the Department of Health. Drs Esene, Karekezi, Khan, Solla and Kolias are members of the Young Neurosurgeons committee of the World Federation of Neurosurgical Societies. The committee is supporting this project.

**Patient consent for publication** Not required.

**Provenance and peer review** Not commissioned; externally peer reviewed.

**ORCID iDs**
Charlotte J Whiffin http://orcid.org/0000-0002-9767-2123

Brandon G Smith http://orcid.org/0000-0001-8471-1368
Tom Bashford http://orcid.org/0000-0003-0228-9779

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
