## [Reviewer comments · BMJ Open]

ARTICLE DETAILS

TITLE (PROVISIONAL)	Neurosurgeons' experiences of conducting and disseminating clinical research in low- and middle-income countries: A qualitative study protocol.
AUTHORS	Whiffin, Charlotte; Smith, Brandon; Esene, Ignatius; Karekezi, Claire; Bashford, Tom; Khan, Muhammad; Davi, Solla; Hutchinson, Peter; Koliass, Angelos

VERSION 1 - REVIEW

REVIEWER	Leonardo C. Welling State University of Ponta Grossa. Paraná/Brazil
REVIEW RETURNED	27-Apr-2020

GENERAL COMMENTS	It is a well-designed and objective study protocol that can help to understand the difficulties that are going on low-and middle-income countries. Despite the limited number of participants, this research maybe will be a pilot project for more studies.
--

REVIEWER	Bertrand DEBONO Department of Neurosurgery Neurosciences Pole Clinique des Cèdres Chateau d'Alliez 31700 Cornebarrieu France
REVIEW RETURNED	27-Apr-2020

GENERAL COMMENTS	The authors propose a qualitative research protocol concerning a well-identified and perfectly noteworthy problem (conducting and disseminating clinical research in low- and middle-income countries). The qualitative methodology presented is adapted to their project, and each step of the presented methodology is suitable and detailed, including rigour and ethics. The organization of online focus groups is a judicious choice regarding the geographical dissemination of the surgeons enrolled in the survey, but I have a question concerning the candidates for the semi-structured interviews of phase 2: 20 surgeons are already planned, but is this number sufficient/adapted upstream, not
--

	knowing when the saturation of the themes will occur? Another short question: who will do the analysis on NVivo? a social science researcher, the UK surgeons, psychologists, public health physicians?... (...although I think there will be the peer-to-peer exchanges that the authors point out). Just to explain the analysis of themes. In conclusion, and in addition to these minor questions, I believe that the subject identified is worthy of research, that the proposed qualitative protocol is well adapted and well designed, that qualitative studies are too rare in neurosurgery and that this project should be accepted.
--	--

VERSION 1 – AUTHOR RESPONSE

Actions required	Comments	Actions	Page
Reviewer: 1			
It is a well-designed and objective study protocol that can help to understand the difficulties that are going on low-and middle-income countries. Despite the limited number of participants, this research maybe will be a pilot project for more studies.	Thank you for this positive feedback	No actions required	
Reviewer: 2			
The authors propose a qualitative research protocol concerning a well-identified and perfectly noteworthy problem (conducting and disseminating clinical research in low-and middle-income countries). The qualitative methodology presented is adapted to their project, and each step of the presented methodology is suitable and detailed, including rigour and ethics.	Thank you for this positive feedback	No actions required	
The organization of online focus groups is a judicious choice regarding the geographical dissemination of the surgeons enrolled in the survey, but I have a question concerning the candidates for the semi-structured interviews of phase 2: 20 surgeons are already planned, but is this number sufficient/adapted upstream, not knowing when the saturation of the themes will occur?	It is very difficult to predict when saturation of themes will occur. However, given that the focus group data will also be used to inform the final findings we are confident a sample of twenty will be sufficient. If, we do not reach saturation we may	We have added to the manuscript that if saturation is not reached, and it is feasible to do so, we will notify the ethics committee of our intention to increase the sample to a more appropriate size.	p.5

	be able to request ethical approval to increase the sample size if this is feasible and appropriate to do so.		
Another short question: who will do the analysis on NVivo? a social science researcher, the UK surgeons, psychologists, public health physicians?... (...although I think there will be the peer-to-peer exchanges that the authors point out). Just to explain the analysis of themes.	NVIVO analysis will be conducted by UK authors C Whiffin and Brandon Smith. Charlotte is an experienced qualitative research and nurse academic Brandon is a PhD student and 4 th year medical student.	We have added a more information to the paper about the role and experience of the authors in data analysis and how/when respondent validation will take place.	p.5